# Internal gene segments from a mouse-adapted influenza B virus confer increased pathogenicity to mice

Arne Matthys[1,2], Laura Amelinck[1,2], Anouk Smet[1,2], Tine Ysenbaert[1,2], Thorsten U. Vogel[3], Xavier Saelens[1,2], João Paulo Portela Catani[1,2]*

**1** Center for Medical Biotechnology, VIB, Ghent, Belgium, **2** Department of Biochemistry and Microbiology, Ghent University, Ghent, Belgium, **3** Sanofi, Research North America, Waltham, Massachusetts, United States of America

* joao.portelacatani@vib-ugent.be

## Abstract

Influenza B viruses (IBVs) contribute significantly to the annual influenza epidemics in human. Most IBV strains are non- or poorly pathogenic in mice, which are frequently used for vaccine studies. We describe the generation of a mouse-adapted IBV strain that retains pathogenicity in mice when carrying hemagglutinin (HA) and neuraminidase (NA) gene segments from a heterologous IBV strain. Serial passage of an influenza B reassortant virus, containing the HA and NA segments from B/Washington/02/2019 on a mouse-adapted B/Memphis/12/1997 backbone, resulted in the selection of an IBV that was highly pathogenic for mice. This mouse-adapted IBV strain had acquired non-synonymous mutations in 5 gene segments. Sequence analysis of the intermediate passages indicated that mutations in the matrix (M), polymerase acidic (PA), and polymerase basic 1 (PB1) gene segments appeared at passages 9 and 13, suggesting that these mutations contributed to the pathogenicity in mice. Mouse challenge studies with rescued reassortant viruses with one or multiple mutated gene segments, confirmed the importance of substitutions in the M and PA segments for pathogenicity. Using the novel mouse-adapted IBV backbone, we rescued reassortant viruses containing the HA and NA segments of B/Austria/1359417/2021 and demonstrated its increased pathogenicity in BALB/c mice compared to IBV rescued on the parental strain. This mouse-adapted IBV backbone provides a valuable tool for the study of IBV in mice.

## Introduction

Influenza B viruses (IBVs) are human respiratory pathogens that cause a significant yearly recurrent disease burden [1]. IBVs diverged in the 1980s into two antigenically distinct lineages, named the B/Victoria/2/1987-like and B/Yamagata/16/1988-like lineage, herein referred as Victoria and Yamagata lineage, respectively. These lineages

**Data availability statement:** All data supporting the findings of this study are included within the paper.

**Funding:** A.M. was supported by a Fonds voor Wetenschappelijk Onderzoek (FWO) PhD fellowship strategic basic research (1S93223N). This work was also supported by Sanofi. No additional external funding was received for this study. The funder provided support in the form of salaries for authors. T.U.V., but did not have any additional role in the study design, data collection and analysis, decision to publish, or preparation of the manuscript.

**Competing interests:** X.S. reports grants from Sanofi and T.U.V. may have stock options from Sanofi. This does not alter our adherence to PLOS ONE policies on sharing data and materials.

have been co-circulating with alternating predominance [2]. Since 2020, however, Yamagata lineage IBVs have not been detected. Given that IBVs lack an established animal reservoir, the continued absence of detectable Yamagata lineage viruses suggests their extinction [3,4].

The segmented, negative-sense RNA genome of IBVs is replicated by the viral RNA-dependent RNA polymerase that lacks proofreading activity, contributing to the genetic variability. Nucleotide substitutions occur in all gene segments and are generally higher for Victoria compared to Yamagata lineage viruses [5]. With $2 \times 10^{-3}$ substitutions/site/year, the IBV hemagglutinin (HA) gene mutates slower than the HA of human H1N1 ($4 \times 10^{-3}$ substitutions/site/year) and H3N2 ($5.5 \times 10^{-3}$ substitutions/site/year) influenza A viruses [6]. Despite the slightly lower substitution rate, antigenic drift in IBV necessitated frequent updates of the recommended vaccine strains (e.g., 6 distinct Victoria or Yamagata strains were recommended in the last 10 years) [7].

IBVs are poorly pathogenic to mice, limiting the use of this small animal model for IBV research. Serial passage of IBV through mouse lungs is a well-established method to select a pathogenic virus. IBV adaptation to mice has been associated with amino acid substitutions across all eight viral gene segments [8–11]. Identification of the acquired mutations, combined with reverse genetics, allows determination of the substitutions that are necessary and sufficient for the mouse-adapted phenotype. Mouse-adapted IBVs have been used to define pathogenic determinants and were exploited for the generation of live attenuated influenza vaccines candidates [12], to study the antibody repertoire and antigenic drift [13], or for drug susceptibility testing [14].

In this study, we report the serial passage of an IBV through mice that resulted in the establishment of a backbone of internal segments that conferred increased mice pathogenicity of IBV reassortants.

## Results

The HA and neuraminidase (NA) segments of B/Washington/02/2019 were used to rescue a 2:6 reassortant virus using the internal segments obtained from the mouse-adapted B/Memphis/12/1997. This initial backbone includes the single substitution (N221S) in BM1, previously shown to confer increased pathogenicity to B/Memphis/12/1997 [11]. Sequence analysis revealed that the rescued reassortant IBV, which was named B/HwNw-Mem97, carried an additional T211A substitution in the HA and M390I in the NA segment. Despite encoding the N221S substitution in the C-terminus of BM1, previously described to enhance mouse virulence, the B/HwNw-Mem97 reassortant virus was not pathogenic to BALB/c mice (Fig 1a). Similarly, B/HwNw-Mem97 was not pathogenic to DBA/2J mice, a mouse strain characterized by increased sensitivity to influenza virus infection (**Fig 1b**) [15,16].

To isolate an IBV strain with enhanced pathogenicity, the B/HwNw-Mem97 virus was passaged multiple times through the lungs of DBA/2J mice and propagated on MDCK cells between each *in vivo* passage. After 13 passages, the infected mice showed clinical signs of disease (ruffled fur and reduced mobility). The virus that was obtained after passage 13, referred to as the B/HwNw-Mem97 m.a. virus, was

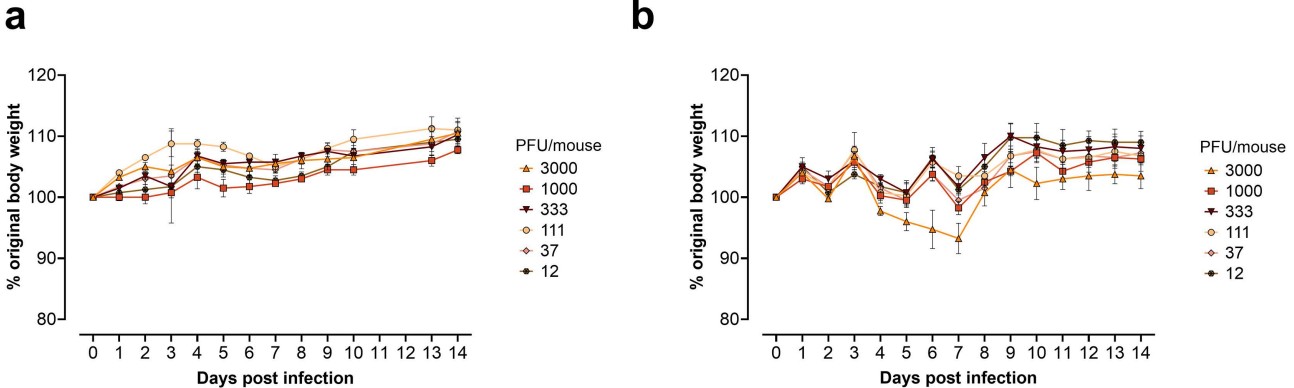

**Fig 1. Weight loss induced by B/HwNw-Mem97.** Different amounts of the 2:6 B/HwNw-Mem97 reassortant with N221S substitution in the C-terminus of BM1 were used to inoculate groups of four BALB/c **(a)** or DBA/2J mice **(b)**. Data points represent the mean relative body weight and error bars the standard error of the mean. The experiment was performed using 4 mice per group.

amplified on MDCK cells and titrated in BALB/c and DBA/2J mice. The resulting 50% lethal dose ($LD_{50}$) were 810 PFU for BALB/c and 25 PFU for DBA/2J mice (**Fig 2**).

To identify the mutations associated with increased pathogenicity, the complete B/HwNw-Mem97 m.a. genome was sequenced. Three synonymous mutations were identified, one in the PB2, one in the PB1, and one in the BM1 segment. In addition, a total of eight non-synonymous mutations were found across PB2, PB1, PA, NA, and BM1 gene segments (**Fig 3a**, **Table 1**). To understand the evolutionary trajectory of each mutation, the segments of intermediate viral passages 3, 6, and 9 were subcloned in the pHW2000 vector and sequenced (**Fig 3b**, **Table 1**).

Reassortant viruses were rescued using combinations of segments from B/HwNw-Mem97 m.a. virus to investigate the individual contributions of each segment to the pathogenicity. These reassortants consisted of combinations of segments derived from the parental B/HwNw-Mem97- and the B/HwNw-Mem97 m.a. virus (**Fig 4a**). Despite multiple attempts, we were unable to rescue B/HwNw-Mem97 m.a. by reverse genetics. We inoculated BALB/c mice with an equal amount of the different reassortant viruses (1620 PFU/mouse, which corresponds to 2 $LD_{50}$ of B/HwNw-Mem97 m.a.) and monitored morbidity and mortality for 14 days (**Fig 4b**). All mutated segments except PB2, or the combination of PB2 and PB1, contributed significantly to morbidity. Mortality was only significantly affected by the combination of mutated segments PB1 and PA, or NA and M. PA and M segments derived from B/HwNw-Mem97 m.a. were predominantly and independently associated with increased mortality. Infection of BALB/c mice with reverse genetics B/HwNw-Mem97 viruses with PB2 segment or with PB2 and PB1 segments derived from B/HwNw-Mem97 m.a., resulted in limited body weight loss (**Fig 4b**).

To assess whether the 6 internal gene segments of B/HwNw-Mem97 m.a. could confer pathogenicity in mice when combined with HA and NA from another IBV, we rescued recombinant viruses carrying the HA and NA gene segments from B/Austria/1359417/2021 and either the six internal segments from B/HwNw-Mem97 m.a. or from B/Memphis/12/1997 m.a. (with the BM1 N221S substitution; [11]). BALB/c mice were inoculated with 450 PFUs of each reassortant. Although mice in both groups experienced body weight loss and mortality, the reassortant virus bearing the B/HwNw-Mem97 m.a. internal genes induced significantly greater weight loss. The mortality outcome did not differ significantly (**Fig 5**).

## Discussion

Aiming to generate an IBV challenge model bearing the HA and NA components of the B/Washington/02/2019 virus, we rescued a 2:6 reassortant strain, which we named B/HwNw-Mem97, with the internal segments of the mouse adapted B/Memphis/12/1997 [11]. The rescued RG B/HwNw-Mem97 was not pathogenic in BALB/c and DBA/2J mice. To generate a

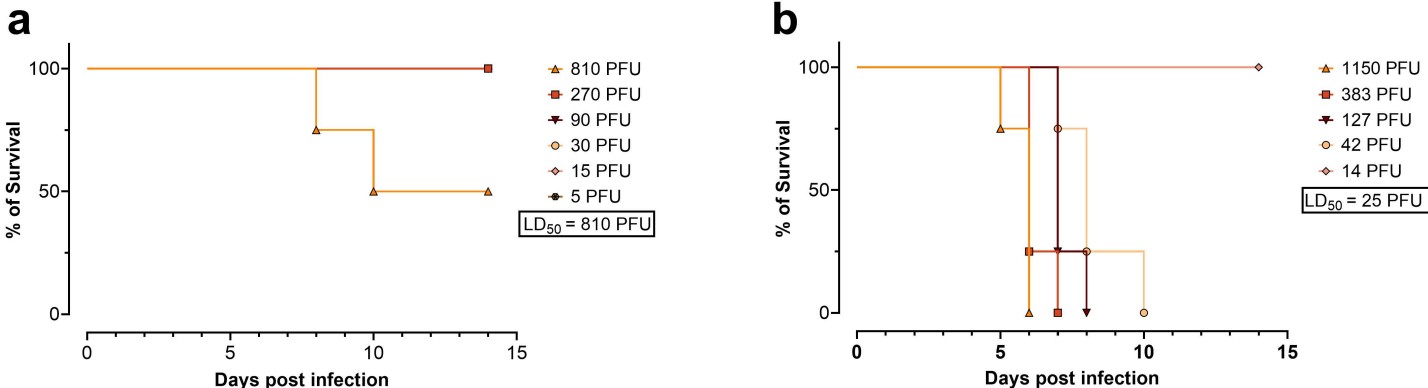

**Fig 2. Survival of mice inoculated with of B/HwNw-Mem97 m.a. virus.** BALB/c (**a**) and DBA/2J mice (**b**) were inoculated with different doses of B/HwNw-Mem97 m.a. virus (obtained after thirteen serial passages in DBA/2J mice). The experiment was performed using 4 mice per group and the $LD_{50}$ was calculated using the Reed and Muench method.

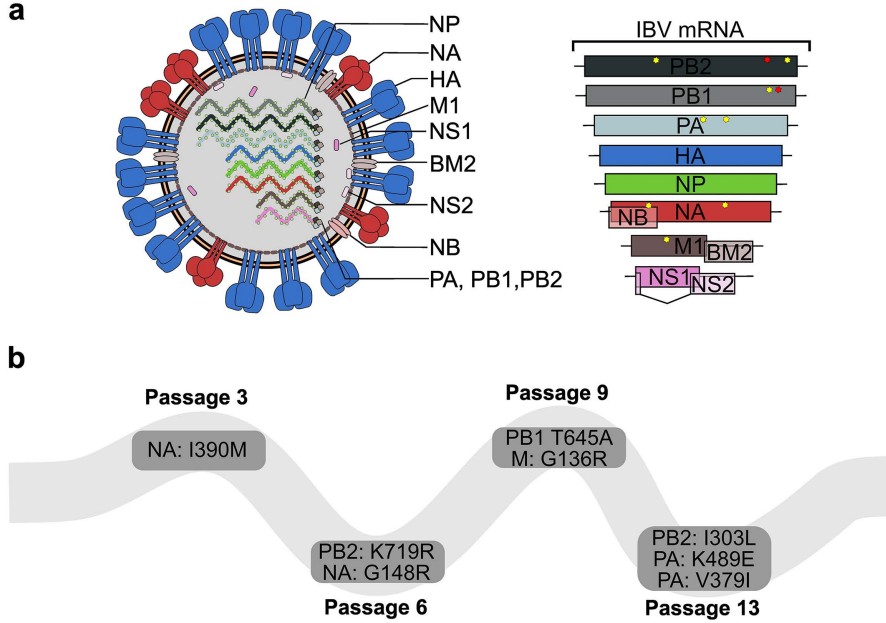

**Fig 3. Multi-segment mouse-adaptation of the B/HwNw-Mem97 m.a. virus. a)** Eight gene segments code for at least 11 IBV proteins. Nucleotide substitutions in IBV after 13 in vivo passages are mapped per segment with yellow or red stars for non-synonymous and synonymous mutations, respectively. **b)** Appearance of mutations resulting in amino acid substitutions present in the B/HwNw-Mem97 m.a. virus, through the tested passages.

pathogenic strain, 13 serial passages were performed in lungs of DBA/2J mice, which are more susceptible to influenza A virus infection than BALB/cByJ mice [17]. This increased susceptibility may explain the difference in LD50 of the B/HwNw-Mem97 m.a. virus in DBA/2J mice (25 PFU) compared to BALB/c mice (810 PFU). The resulting mouse adapted IBV, named B/HwNw-Mem97 m.a., had acquired non-synonymous mutations in the PB2, PB1, PA, NA and BM. Among these, the substitutions in the M and PA segments played a predominant role in increasing mice pathogenicity.

A substitution at the C-terminus of the M segment (N221S) has previously been described as a determinant of IBV mouse pathogenicity [11]. Although this substitution was present in our rescued B/HwNw-Mem97 virus, this virus did

**Table 1. Mutations present in the B/HwNw-Mem97 m.a. genome relative to the parental rescued virus. Mutations were identified in intermediate passages (P3, P6, or P9) with Sanger sequencing and in the mouse adapted virus (P13) with NGS and Sanger sequencing.**

| Passage | Viral segment (protein)[1] | Nucleotide change (codon)[2] | Amino acid change[4,5] |
|---|---|---|---|
| P3 | 6 (NA/NB) | 1170 (ATA to ATG) | 390 (I to M) |
| P6 | 1 (PB2) | 1929 (TTG to TTA) | – |
| | | 2171 (AAG to AGG) | 719 (K to R) |
| | 6 (NA/NB) | 481 (GGA to AGA) | 148 (G to R) |
| | | 1170 (ATA to ATG) | 390 (I to M) |
| | 2 (PB1) | 1952 (ACC to GCC) | 645 (T to A) |
| | | 2134 (AGG to AGA) | – |
| P9 | 6 (NA/NB) | 961 (GAG to AAG)[3] | 308 (E to K) |
| | | 1170 (ATA to ATG) | 390 (I to M) |
| | | 1345 (GAG to AGA)[3] | 436 (E to K) |
| | | 1474 (TGT to TAT)[3] | – |
| | 7 (BM1/BM2) | 411 (TAC to TAT) | – |
| | | 421 (GGA to AGA) | 136 (G to R) |
| P13 (m.a.) | 1 (PB2) | 937 (ATA to CTA) | 303 (I to L) |
| | | 1938 (TTG to TTA) | – |
| | | 2180 (A to G) | 719 (K to R) |
| | 2 (PB1) | 1952 (ACC to GCC) | 645 (T to A) |
| | | 2134 (AGG to AGA) | – |
| | 3 (PA) | 1135 (GTA to ATA) | 379 (V to I) |
| | | 1465 (AAA to GAA) | 489 (K to E) |
| | 6 (NA/NB) | 481 (GGA to AGA) | 148 (G to R) |
| | | 1170 (ATA to ATG) | 390 (I to M) |
| | 7 (BM1/BM2) | 411 (TAC to TAT) | – |
| | | 421 (GGA to AGA) | 136 (G to R) |

[1]Segments 1 (PB2), 3 (PA), 5 (NP), and 8 (NS1/NS2) were not analysed for passage 9.

[2]Nucleotide numbering relative to the reference genome (see Materials and Methods).

[3]Mutation detected in at least 50% of the sequenced samples.

[4]"-" indicates synonymous mutation

[5]Amino acid numbering relative to reference proteins (see Materials and Methods).

not cause detectable morbidity in mice. After 9 serial passages, the substitution G136R was identified in the M segment and was found to correlate with increased mouse pathogenicity. Substitutions in the M segment have also been linked to increased pathogenicity in mice of influenza A viruses, however the mechanism remains unclear and may involve altered vRNP binding [18].

In the PA segment, a single substitution (K338R) has been reported to enhance pathogenicity in mice by approximately 10-fold across both IBV lineages, and this effect was associated with increased replicase activity [10]. However, pathogenicity still required relatively high viral doses ($LD_{50}$ of $10^{3.5}$ PFU and $10^{5.5}$ PFU for B/Victoria- and B/Yamagata-lineage viruses, respectively). The B/HwNw-Mem97 m.a. PA bears two substitutions (V379I and K489E). Interestingly, high-yielding egg-adapted IBV backbones were recently described and include the PA substitution K489E [19].

Adaptation of influenza viruses to mice can result in substitutions in HA and NA, which may be undesirable when studying immune responses targeting these antigens. For example, seventeen serial passages in mice of a B/Victoria clade 1A virus (B/Novosibirsk/40/2017) resulted in substitutions in HA (T214I) and NA (D432N) [9]. Seventeen passages of B/Florida/04/2006 resulted in 5 amino acid changes, of which one in HA (D424G) [20].

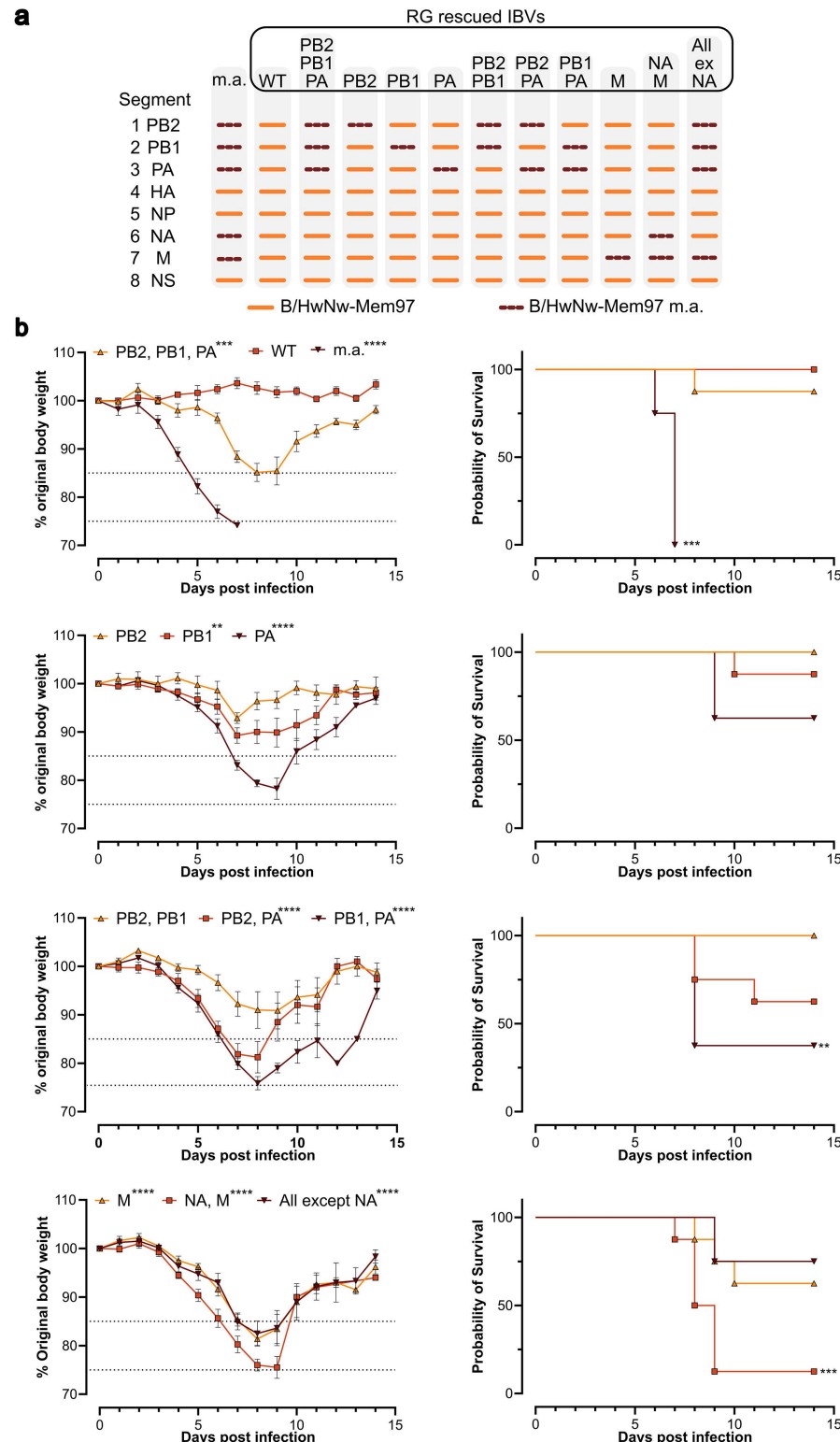

**Fig 4. Morbidity and mortality in BALB/c mice challenged with different B/HwNw-Mem97 m.a. reassortant viruses. a)** Genomic composition of the B/HwNw-Mem97 m.a. (m.a.) and rescued reverse genetics (RG) IBV viruses. Note that the sequences of the HA, NP, and NS segments of B/HwNw-Mem97 and B/HwNw-Mem97 m.a. are identical. **b)** Bodyweight (left) and survival curves (right) of BALB/c mice over time following challenge with 1620

PFU/mouse. Differences in bodyweight between groups (compared to the WT group, shown only in the first graph) were tested by two-way ANOVA with Dunnett's multiple comparison: **p = 0.0021, ***P < 0.0002, ****P < 0.0001. Differences in survival (compared to the WT group) were tested with a log-rank (Mantel-Cox) test: **p = 0.0021, ***p < 0.0002, ****p < 0.0001. The graphs show compiled data from two independent experiments (n = 4 mice per group per experiment). The dotted lines in the bodyweight graphs indicate the ethical maximum permitted bodyweight reduction.

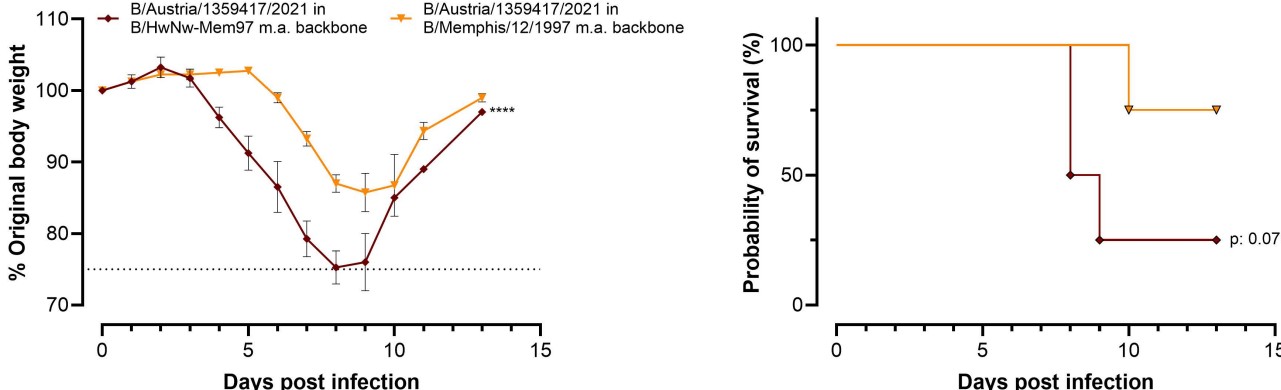

**Fig 5. The internal gene segments of B/HwNw-Mem97 m.a. confer increased pathogenicity of RG B/Austria/1359417/2021 virus to BALB/c mice.** Bodyweight and survival curves of female BALB/c mice over time following challenge with 450 PFU/mouse (n = 4 per group). Differences in bodyweight between groups were tested by two-way ANOVA with Dunnett's multiple comparison: ****p < 0.0001. Differences in survival were tested with a log-rank (Mantel-Cox). The dotted lines in the bodyweight graphs indicate the ethical maximum permitted bodyweight reduction.

The B/HwNw-Mem97 m.a. virus as reported here carries the HA substitution T211A and NA substitutions G148R and I390M. Notable, the T211A in HA and M390I in NA were present in the rescued parental strain (P0), and consequently occurred independently of the mouse adaptation. The methionine at position 390 in NA is highly conserved among IBVs. Although the rescued B/HwNw-Mem97 m.a. initially carried this M390I substitution, it was reverted back to a methionine (I390M) after, at the most, three passages. Thus, given that the NA segment at least partially contributed to pathogenicity in mice, it is likely that the G148R substitution in NA is associated with increased pathogenicity. Typically, position 148 in NA features a glycine in Victoria lineage viruses or a glutamic acid in Yamagata lineage viruses. Neighbouring residues (especially D149) are critical components of the catalytic site, directly participating in sialic acid receptor binding [21]. Substitution G148R may introduce local allosteric changes with a limited contribution to viral pathogenicity, since reassortant viruses with the parental NA segment were also pathogenic in mice.

The rescue of the 2:6 B/Austria/1359417/2021 reassortant on our B/HwNw-Mem97 m.a. backbone resulted in a virus with increased pathogenicity in comparison to its counterpart rescued on the B/Memphis/12/1997 mouse adapted backbone. No substitutions were found in the HA or NA derived from B/Austria/1359417/2021 in the reassortant virus, indicating that mouse pathogenicity relied solely on the m.a. internal segments, allowing the study in mice of IBV viruses with potentially unaltered antigenicity.

Our study has limitations. First, although we identified new key substitutions in the PA and M segments, the precise molecular mechanism underlying the increased pathogenicity remains to be investigated. Second, we did not assess the cytokine profile or lung histopathology, important parameters to understand the mechanism linked to disease severity [22–24].

In summary, we isolated an IBV backbone carrying non-synonymous substitution in 5 gene segments (PB2, PB1, PA, NA, and M). Except for PB2, all the mutated segments seem to, at least partially, contribute to increased mice

pathogenicity. The m.a. backbone was successfully used to rescue a second virus comprising HA and NA of the recent B/Austria/1359417/2021 vaccine strain. The described backbone offers a new platform to rescue IBV strains that are pathogenic in mice, without altering their surface antigenicity.

## Materials and methods

### Virus rescue

HA and NA sequences from B/Washington/02/2019 (GISAID accession numbers EPI1368874 and EPI1368872, respectively) and B/Austria/13595417/2021 (GISAID accession numbers EPI1845793 and EPI1845794, respectively) were generated by custom gene synthesis (GeneArt, Thermo Fischer) and cloned in the bidirectional expression plasmid pHW2000. The gene segments from the mouse adapted B/Memphis/12/1997 [11] were reverse transcribed from the virus stock, using the primers described previously [25] and cloned in the pHW2000 expression vector. To rescue reassortant viruses, a co-culture of MDCK and HEK293 cells was transfected with 1 µg of plasmids encoding HA and NA of B/Washington/02/2019, and the 6 plasmids (PB2, PB1, PA, NP M, and NS) encoding internal proteins of the mouse adapted B/Memphis/12/1997. The rescue using all 8 parental B/Memphis/12/97 segments was used as positive control, and the negative control consisted of the same plasmids, except for the one encoding NP. The supernatant (SN) was recovered after observation of cytopathic effect and titrated (PFU determination). All rescued viruses were amplified on MDCK cells to prepare working stocks. Briefly, MDCK cells were inoculated with a MOI of 0.001 in the presence of TPCK-treated trypsin (2 µg/mL, Sigma T1426). After cytopathic effect was evident, the cell culture supernatant was recovered, cleared from cells and debris by centrifugation, and titrated by plaque assay on MDCK cells (PFU determination).

### Mouse adaptations

The initial viral infection was performed with 1 x $10^5$ PFU (50 µL per nostril). During each passage, groups of 3 DBA/2J mice were inoculated and euthanized 72 hr later to recover virus from the lungs. Lung homogenates were prepared in PBS using sterile metal beads and a Tissue Lyser II mechanical homogenizer. The homogenates were clarified by centrifugation at 1,000 × g for 10 minutes at 4°C, then pooled and diluted 1:1000 prior to inoculation of MDCK cells. Infected cells were maintained in the presence of 2 µg/mL TPCK-treated trypsin (Sigma-Aldrich, T1426) until observation of cytopathic effect. The cell culture medium was then recovered, cleared by centrifugation and used at 1/100 dilution to inoculate naive mice. These passages were repeated 13 times when clinical signs of influenza virus infection in mice (ruffled fur and reduced mobility) was observed on day 3 after inoculation.

### Virus sequence analysis

The mouse-adapted and parental strains were sequenced using Oxford nanopore at PathoSense (Belgium). The internal genes of intermediate passages were reverse transcribed using the primers described by [26], and the amplified segments were cloned into the bidirectional plasmid pHW2000 and sequenced by Sanger sequencing (IDT Belgium). Nucleotide substitutions are numbered relative to the B/Washington/02/2019 reference genome retrieved from the NCBI viral genomes resource database [27] with the following accession codes for HA (MK676294) and NA (MK676296). Nucleotide substitutions in the internal gene segments are numbered relative to the B/Memphis/12/1997 reference genome with following accession codes for PB1 (AY260942), PB2 (AY260943), PA (AY260944), NP (AY260946), M (AY260941), and NS (AY260948).

### Mouse housing and viral challenge

Specific pathogen-free female BALB/c and DBA/2J mice were obtained from Janvier (France). The animals were housed in a temperature-controlled environment with 12 h light/dark cycles; food and water were provided ad libitum. The animal

facility operates under the Flemish Government License Number LA1400563. All experiments were done under conditions specified by law and authorized by the Institutional Ethical Committee on Experimental Animals (Ethical Application EC2023−120 and EC2023−140). Intranasal administration of virus dilution (50 μL per nostril) was performed under isoflurane sedation in a BSL2 graded facility. Mice were monitored daily for two weeks for weight loss and clinical signs of disease, including inactivity, ruffled fur, labored breathing, and huddling behavior. Animals that lost more than 25% of their original body weight or exhibited reduced mobility or distress associated with labored breathing (indicative of pneumonia) were euthanized immediately by cervical dislocation. No spontaneous deaths were observed. Researchers responsible for the mouse experiments were certified in accordance with the Belgian Royal Decree of 29 May 2013. The $LD_{50}$ was calculated using the Reed and Muench method [28].

## Author contributions

**Conceptualization:** Thorsten U. Vogel, Xavier Saelens, João Paulo Portela Catani.

**Data curation:** João Paulo Portela Catani.

**Formal analysis:** João Paulo Portela Catani.

**Funding acquisition:** Thorsten U. Vogel, Xavier Saelens, João Paulo Portela Catani.

**Investigation:** Laura Amelinck, Anouk Smet, Tine Ysenbaert, João Paulo Portela Catani.

**Methodology:** João Paulo Portela Catani.

**Supervision:** Thorsten U. Vogel, Xavier Saelens, João Paulo Portela Catani.

**Visualization:** Arne Matthys, Laura Amelinck, Anouk Smet.

**Writing – original draft:** Arne Matthys, João Paulo Portela Catani.

**Writing – review & editing:** Arne Matthys, Laura Amelinck, Anouk Smet, Thorsten U. Vogel, Xavier Saelens, João Paulo Portela Catani.

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
