## [Decision Letter · Decision Letter 0]

9 Dec 2025

Dear Dr. Portela Catani,

Thank you for submitting your manuscript to PLOS ONE. After careful consideration, we feel that it has merit but does not fully meet PLOS ONE’s publication criteria as it currently stands. Therefore, we invite you to submit a revised version of the manuscript that addresses the points raised during the review process.

During the revision process, please address all reviewer comments while paying close attention to those associated with experimental design and the inclusion/justification of appropriate controls.

We look forward to receiving your revised manuscript.

Kind regards,

Victor C Huber

Academic Editor

PLOS One

Journal Requirements:

2. To comply with PLOS One submissions requirements, in your Methods section, please provide additional information regarding the experiments involving animals and ensure you have included details on methods of anesthesia and/or analgesia, and efforts to alleviate suffering.

[A.M. was supported by a FWO PhD fellowship strategic basic research (1S93223N). This work was also supported by Sanofi.].

5. Please expand the acronym “FWO” (as indicated in your financial disclosure) so that it states the name of your funders in full.

6. Thank you for stating the following in the Financial Disclosure section:

[A.M. was supported by a FWO PhD fellowship strategic basic research (1S93223N). This work was also supported by Sanofi.].

We note that you received funding from a commercial source: Sanofi.

7. Thank you for stating the following in the Competing Interests section:

[X.S. reports grants from Sanofi and T.U.V. may have stock options from Sanofi.].

We note that one or more of the authors have an affiliation and/or employed to the commercial funders of this research study: Sanofi.

8. Your ethics statement should only appear in the Methods section of your manuscript. If your ethics statement is written in any section besides the Methods, please move it to the Methods section and delete it from any other section. Please ensure that your ethics statement is included in your manuscript, as the ethics statement entered into the online submission form will not be published alongside your manuscript.

Reviewers' comments:

Reviewer's Responses to Questions

**Comments to the Author**

1. Is the manuscript technically sound, and do the data support the conclusions?

Reviewer #1: Yes

Reviewer #2: Yes

2. Has the statistical analysis been performed appropriately and rigorously?

Reviewer #1: Yes

Reviewer #2: Yes

3. Have the authors made all data underlying the findings in their manuscript fully available?

Reviewer #1: Yes

Reviewer #2: Yes

4. Is the manuscript presented in an intelligible fashion and written in standard English?

Reviewer #1: Yes

Reviewer #2: Yes

Reviewer #1: Matthys and colleagues developed a mouse-adapted IBV strain based on a previously published B/Memphis/12/1997 backbone. They assessed how acquisition of mutations in all viral segments contribute to enhanced pathogenicity in mice, and rescued an reassortant IBV strain which contained the most recent B/Austria/2021 HA and NA.

Given the low pathogenicity of IBV strains in mice in general, this is a useful tool which can be used to study IBV pathogenesis in vivo.

I have a few minor comments which I hope will improve the quality of the manuscript:

- Line 33: add complete lineage names, e.g. B/Victoria/2/1987-like (B/Victoria) and B/Yamagata/16/1988-like

- Line 36/37: there is an issue with the bibliography (numbers vs last authors name in some citations)

- Line 45: is the 6 IBV strains in the last 10 years in Victoria only? If so, please comment on this.

- Line 71 ff (Figure 1 legend): were four or three mice used per group? This is currently unclear. Also more appropriate to speak of 'weight change over time' instead of 'weight loss'.

- Line 83 ff (Figure 2 legend): Change pathogenicity to something like 'Kaplan Meier survival curve'. Pathogenicity could also be reflected in weight loss, viral replication across respiratory organs etc, and this was not assessed here.

- Line 122 ff (Figure 4). The figure and the used colours are confusing. As said in Line 124, the HA, NP and NS segments are identical between the Mem97 WT and Mem97 m.a., then why are the colours different? I think it would be helpful to split the figure up and first show what segments are generated and how they differ between the Mem97 WT and the MEM97 m.a., and then show in a subsequent figure what RG viruses have been generated and tested.

Line135 ff: I find the usage of the word 'heterologous' confusing in this context, because the B/HwNw-Mem97 is already per se an heterologous virus, as it contains the interal segments from B/Mem/1997 and HA/NA from B/Washington/2019.

Reviewer #2: The manuscript addresses an important topic in influenza B virus research by generating mouse-adapted viruses and examining how different viral gene segments contribute to pathogenicity. The study is thoughtfully designed, and the combination of reverse genetics, sequencing, and mouse infection models offers useful insights into virulence determinants. The experiments appear carefully performed, and the use of multiple reassortant viruses adds strength to the work.

That said, several areas need clarification or additional detail to improve transparency and reproducibility.

For the virus that could not be rescued (B/HwNw-Mem97 m.a.), please provide a hypothesis for why rescue failed and discuss possible technical or biological reasons.

PFU measurements were used to monitor virus rescue and replication, and the rescued viruses were confirmed by sequencing. However, no independent positive control was included. Please discuss this limitation and explain how sequencing combined with PFU measurements ensured that the rescue system worked reliably.

No negative control (such as mock-transfected cells or PBS-treated mice) was included. Please clarify how background effects or potential contamination were ruled out.

Please clarify whether PFU or viral titers were measured after each mouse passage during adaptation, or whether cytopathic effect (CPE) was the only readout. Quantitative titers across passages would strengthen conclusions about viral adaptation.

Were lung viral titers measured during the adaptation passages or in the BALB/c and DBA/2J infection studies? If not, please consider adding these data, as they would support the morbidity and mortality observations.

Were histopathological analyses performed on lung tissue to confirm pathology and correlate with clinical signs? If not, please discuss this as a limitation.

Were cytokine profiles or other immune markers measured to provide mechanistic insight into disease severity? If feasible, please consider including these data or discuss this limitation.

Please clarify the rationale for choosing DBA/2J mice for adaptation and BALB/c mice for subsequent infection studies, and discuss how strain-specific susceptibility might influence interpretation.

The manuscript states that virulence is independent of HA and NA. Please expand on the mechanism and explain how the data support this conclusion.

Finally, please include a more explicit discussion of the study’s limitations.

**Do you want your identity to be public for this peer review?** For information about this choice, including consent withdrawal, please see our Privacy Policy

Reviewer #1: No

Reviewer #2: No

---

## [Author Response · Author response to Decision Letter 1]

28 Jan 2026

Ref. [PONE-D-25-54336] - [EMID:03d00e8efc403f93] - Multi-segment mouse-adaptation of a recent B/Victoria-lineage virus independent from substitutions in hemagglutinin and neuraminidase

We are grateful to the reviewers for their time and valuable comments on our manuscript. Below, please find our point-by-point responses (in blue font) to the comments that were raised.

Reviewer #1: Matthys and colleagues developed a mouse-adapted IBV strain based on a previously published B/Memphis/12/1997 backbone. They assessed how acquisition of mutations in all viral segments contribute to enhanced pathogenicity in mice, and rescued an reassortant IBV strain which contained the most recent B/Austria/2021 HA and NA.

Given the low pathogenicity of IBV strains in mice in general, this is a useful tool which can be used to study IBV pathogenesis in vivo.

I have a few minor comments which I hope will improve the quality of the manuscript:

- Line 33: add complete lineage names, e.g. B/Victoria/2/1987-like (B/Victoria) and B/Yamagata/16/1988-like

Thank you for the valuable remark. Complete lineage names were added

- Line 36/37: there is an issue with the bibliography (numbers vs last authors name in some citations)

Thank you for the meticulous review, the bibliography was corrected.

- Line 45: is the 6 IBV strains in the last 10 years in Victoria only? If so, please comment on this.

We refer to both strains, which we clarified in the revised manuscript as follows: “... (e.g. 6 distinct Victoria or Yamagata strains were recommended in the last 10 years)...”

- Line 71 ff (Figure 1 legend): were four or three mice used per group? This is currently unclear. Also more appropriate to speak of 'weight change over time' instead of 'weight loss'.

Thank you for highlighting this. We now specify in the figure legend that four mice were used per group. In addition, the caption of the Y-axis in the graphs now reads “% original body weight”.

- Line 83 ff (Figure 2 legend): Change pathogenicity to something like 'Kaplan Meier survival curve'. Pathogenicity could also be reflected in weight loss, viral replication across respiratory organs etc, and this was not assessed here.

“Pathogenicity” was replaced by: “Survival of mice inoculated with...”

- Line 122 ff (Figure 4). The figure and the used colours are confusing. As said in Line 124, the HA, NP and NS segments are identical between the Mem97 WT and Mem97 m.a., then why are the colours different? I think it would be helpful to split the figure up and first show what segments are generated and how they differ between the Mem97 WT and the MEM97 m.a., and then show in a subsequent figure what RG viruses have been generated and tested.

Thank you for this comment. All the segments were amplified from B/HwNw-Mem97 m.a. virus, and then used to rescue the reassortants as depicted in the figure. That is the reason why they were colored differently. However, we agree that color distinction is not necessary for the three non mutated segments. The figure was adapted accordingly.

Line135 ff: I find the usage of the word 'heterologous' confusing in this context, because the B/HwNw-Mem97 is already per se an heterologous virus, as it contains the interal segments from B/Mem/1997 and HA/NA from B/Washington/2019.

We agree with this comment. To improve clarity our statement was rephrased as follows: “To assess whether the 6 internal gene segments of B/HwNw-Mem97 m.a. could confer pathogenicity in mice when combined with HA and NA from another IBV, we rescued recombinant viruses carrying the HA and NA gene segments from B/Austria/1359417/2021 and either the six internal segments from B/HwNw-Mem97 m.a. or from B/Memphis/12/1997 m.a.”

Reviewer #2: The manuscript addresses an important topic in influenza B virus research by generating mouse-adapted viruses and examining how different viral gene segments contribute to pathogenicity. The study is thoughtfully designed, and the combination of reverse genetics, sequencing, and mouse infection models offers useful insights into virulence determinants. The experiments appear carefully performed, and the use of multiple reassortant viruses adds strength to the work.

That said, several areas need clarification or additional detail to improve transparency and reproducibility.

For the virus that could not be rescued (B/HwNw-Mem97 m.a.), please provide a hypothesis for why rescue failed and discuss possible technical or biological reasons.

We believe that the failure to rescue the B/HwNw-Mem97 m.a. is technical. The major argument for that is the successful rescue of an IBV carrying the HA and NA gene segments from B/Austria/1359417/2021 and the six internal segments from B/HwNw-Mem97 m.a..

PFU measurements were used to monitor virus rescue and replication, and the rescued viruses were confirmed by sequencing. However, no independent positive control was included. Please discuss this limitation and explain how sequencing combined with PFU measurements ensured that the rescue system worked reliably.

Please note that a positive control was included in the rescue: B/Memphis/12/97. The positive control was used to confirm cytopathic effect in HEK/MDCK co-cultures but was not stored or sequenced. This information was not included in Methods section:

“The rescue using all 8 parental B/Memphis/12/97 segments was used as positive control, and the negative control consisted of the same plasmids, except for the one encoding NP”

No negative control (such as mock-transfected cells or PBS-treated mice) was included. Please clarify how background effects or potential contamination were ruled out.

A negative control was included in rescue and comprised a transfection of all segments except NP and no cytopathic effect was observed in this setup.

Please clarify whether PFU or viral titers were measured after each mouse passage during adaptation, or whether cytopathic effect (CPE) was the only readout. Quantitative titers across passages would strengthen conclusions about viral adaptation.

The titration of mouse passages was not included at the time of the mouse adaptation experiments. Our set up allowed to perform one passage per week, so 13 weeks in total. If done sequentially, the titration would extend this period to at least 26 weeks. We have, however, now titrated individually and pooled lung homogenates of passage 3, 6, 9, and 12. The figure below suggests that there is no consistent increase in lung viral titers with increased passage number.

Were lung viral titers measured during the adaptation passages or in the BALB/c and DBA/2J infection studies? If not, please consider adding these data, as they would support the morbidity and mortality observations.

Thank you for the suggestion. However, lung virus titers were not measured during the adaptation passage or in the BALB/c and DBA/2J mice. Please see also our response to the previous comment.

Were histopathological analyses performed on lung tissue to confirm pathology and correlate with clinical signs? If not, please discuss this as a limitation.

Thank you for this remark. We have amended our discussion by including a statement on limitations of our work:

“Our study has limitations. First, although we identified new key substitution in the PA and M segments, the precise molecular mechanism underlying the increased pathogenicity remains to be investigated. Second, we did not assess the cytokine profile or lung histopathology, important parameters to understand the mechanism linked to disease severity.”

Were cytokine profiles or other immune markers measured to provide mechanistic insight into disease severity? If feasible, please consider including these data or discuss this limitation.

Please see response to the previous answer.

Please clarify the rationale for choosing DBA/2J mice for adaptation and BALB/c mice for subsequent infection studies and discuss how strain-specific susceptibility might influence interpretation.

DBA/2J mice were described to be more susceptible to influenza (10.1371/journal.pone.0004857), a phenotype that is attributed to a dysfunction of alveolar macrophages and increased susceptibility to infection of the airways (10.1089/jir.2014.0237). DBA/2J mice also have other dysfunctionality in immune system such as a lack of NKG2A receptors (10.1007/s00251-018-01100-x) and C5 complement (10.1016/S0021-9258(19)39817-5). BALB/c mice are very often used to study influenza virus infection, which is why we turned to BALB/c mice to further evaluate the contribution to pathogenicity of reassortant IBVs with the 6-segment mouse-adapted backbone.

In the discussion, we added the following to address this comment:

“Mouse adaptation was performed by serial passages in DBA/2J mice, which are more susceptible to influenza A virus infection than BALB/cByJ mice (10.1371/journal.pone.0004857 and 10.1371/journal.pone.0004857). This increased susceptibility may explain the difference in LD50 of the B/HwNw-Mem97 m.a. virus in DBA/2J mice (25 PFU) compared to BALB/c mice (810 PFU).”

The manuscript states that virulence is independent of HA and NA. Please expand on the mechanism and explain how the data support this conclusion.

Thank you for this remark. We realize that the original title of our manuscript was somewhat misleading. We meant to convey that we were able to generate increased pathogenic virus using internal gene segment from a mouse-adapted IBV combined with HA and NA derived from recent IBV strains. To address this remark, we adapted the title of our manuscript as follows:

“Internal gene segments from a mouse-adapted influenza B virus confer increased pathogenicity to mice”

Finally, please include a more explicit discussion of the study’s limitations.

We added the following statement on limitations of our study to the discussion:

“Our study has limitations. First, although we identified new key substitution in the PA and M segments, the precise molecular mechanism underlying the increased pathogenicity remains to be investigated. Second, we did not assess the cytokine profile or lung histopathology, important parameters to understand the mechanism linked to disease severity.”

---

## [Decision Letter · Decision Letter 1]

2 Mar 2026

Internal gene segments from a mouse-adapted influenza B virus confer increased pathogenicity to mice

PONE-D-25-54336R1

Dear Dr. Portela Catani,

We’re pleased to inform you that your manuscript has been judged scientifically suitable for publication and will be formally accepted for publication once it meets all outstanding technical requirements.

Kind regards,

Victor C Huber

Academic Editor

PLOS One

Additional Editor Comments (optional):

Reviewers' comments:

Reviewer's Responses to Questions

**Comments to the Author**

Reviewer #1: All comments have been addressed

Reviewer #2: All comments have been addressed

2. Is the manuscript technically sound, and do the data support the conclusions?

Reviewer #1: (No Response)

Reviewer #2: Yes

3. Has the statistical analysis been performed appropriately and rigorously?

Reviewer #1: (No Response)

Reviewer #2: Yes

4. Have the authors made all data underlying the findings in their manuscript fully available?

Reviewer #1: (No Response)

Reviewer #2: Yes

5. Is the manuscript presented in an intelligible fashion and written in standard English?

Reviewer #1: (No Response)

Reviewer #2: Yes

Reviewer #1: (No Response)

Reviewer #2: Thank you for the careful revision and for providing detailed responses to the reviewer’s comments. The manuscript has improved in clarity and transparency following revision.

The clarification of positive and negative controls in the reverse genetics rescue system strengthens confidence in the experimental approach. The explanation regarding the unsuccessful rescue of B/HwNw-Mem97 m.a. is reasonable, particularly in light of the successful rescue of reassortant viruses sharing the same internal backbone. The addition of viral titration data across selected mouse passages provides helpful context for the adaptation process.

The rationale for the use of DBA/2J mice for adaptation and BALB/c mice for infection studies is now clearly justified and appropriately supported by literature. Importantly, the revision of the manuscript title and clarification of the role of internal gene segments address concerns regarding potential overinterpretation of HA and NA independence.

Although measurements of lung viral titers during infection, histopathological analysis, and immune profiling would further strengthen mechanistic insight, the authors have appropriately acknowledged these as limitations in the discussion.

Overall, the revisions satisfactorily address the reviewer’s concerns and improve the manuscript.

**Do you want your identity to be public for this peer review?** For information about this choice, including consent withdrawal, please see our Privacy Policy

Reviewer #1: No

Reviewer #2: **Yes:** Mahmuda Yeasmin

---

## [Editor Report · Acceptance letter]

PONE-D-25-54336R1

PLOS One

Dear Dr. Portela Catani,

I'm pleased to inform you that your manuscript has been deemed suitable for publication in PLOS One. Congratulations! Your manuscript is now being handed over to our production team.

Kind regards,

on behalf of

Dr. Victor C Huber

Academic Editor

PLOS One